# Investigation on the Performance of CO_2_ Absorption in Ceramic Hollow-Fiber Gas/Liquid Membrane Contactors

**DOI:** 10.3390/membranes13020249

**Published:** 2023-02-19

**Authors:** Chii-Dong Ho, Hsuan Chang, Yu-Han Chen, Thiam Leng Chew, Jui-Wei Ke

**Affiliations:** 1Department of Chemical and Materials Engineering, Tamkang University, New Taipei 251301, Taiwan; 2Department of Chemical Engineering, Faculty of Engineering, Universiti Teknologi PETRONAS, Seri Iskandar 32610, Malaysia; 3CO2 Research Center (CO2RES), Institute of Contaminant Management, Universiti Teknologi PETRONAS, Seri Iskandar 32610, Malaysia

**Keywords:** carbon dioxide absorption, MEA absorbent, Happel’s free surface model, Sherwood number, ceramic hollow-fiber membrane contactor

## Abstract

The absorption efficiencies of CO_2_ in ceramic hollow-fiber membrane contactors using monoethanolamine (MEA) absorbent under both cocurrent- and countercurrent-flow operations were investigated theoretically and experimentally; various MEA absorbent flow rates, CO_2_ feed flow rates, and inlet CO_2_ concentrations were used as parameters. Theoretical predictions of the CO_2_ absorption flux were analyzed by developing the mathematical formulations based on Happel’s free surface model in terms of mass transfer resistances in series. The experiments of the CO_2_ absorption were conducted by using alumina (Al_2_O_3_) hollow-fiber membranes to confirm the accuracy of the theoretical predictions. The simplified expression of the Sherwood number was formulated to calculate the mass transfer coefficient of the CO_2_ absorption incorporating experimental data. The data were obtained numerically using the fourth-order Runge–Kutta method to predict the concentration distribution and absorption rate enhancement under various fiber packing configurations accomplished by the CO_2_/N_2_ stream passing through the fiber cells. The operations of the hollow-fiber membrane contactor encapsulating *N* = 7 fiber cells and *N* = 19 fiber cells of different packing densities were fabricated in this work to examine the device performance. The accuracy derivation between experimental results and theoretical predictions for cocurrent- and countercurrent-flow operations were $1.31\times 10^{-2}\leq E\leq 4.35\times 10^{-2}$ and $3.90\times 10^{-3}\leq E\leq 2.43\times 10^{-2}$, respectively. A maximum of 965.5% CO_2_ absorption rate enhancement was found in the module with embedding multiple fiber cells compared with that in the device with inserting single-fiber cell. Implementing more fiber cells offers an inexpensive method of improving the absorption efficiency, and thus the operations of the ceramic hollow-fiber membrane contactor with implementing more fiber cells propose a low-priced design to improve the absorption rate enhancement. The higher overall CO_2_ absorption rate was achieved in countercurrent-flow operations than that in cocurrent-flow operations.

## 1. Introduction

Flue gases from fossil fuel combustion contain CO_2_, and as the major contributor of greenhouse effect and climate change, they have attracted much more attention than ever before all over the world [1]. All absorption applications (physical or chemical) are the most common purification technology for CO_2_ removal, which aims to find the solvent formulation to reach the lowest possible energy consumption in environmentally friendly and stable processes [2,3]. Membrane processes have been widely applied to gas absorption and metal ion removal due to their low energy consumption [4]. The membrane gas absorption, which combines the merits of membrane separation and chemical absorption, is widely and commonly used. A mature approach provides higher mass transfer rate and larger gas–liquid contacting area [5,6] compared with conventional absorption methods [7,8]. Since its simplicity overcomes the operational limitations for continuous operations and modulation arrangement, it has promising large-scale industry implementation [9,10]. The application of the membrane contactor to the CO_2_ absorption process is the gas mixture initially diffusing through the gas/liquid interface on both membrane surfaces with the occurring chemical reaction. Then, CO_2_ reacts with the liquid at the membrane pores [11,12,13]. In a microporous hydrophobic membrane contactor, the gas mixture flows on one side, while the absorbent always flows on the other side directly contacting the membrane surface [14].

The most commonly used hollow-fiber membrane contactors were first investigated by Qi and Cussler [15], which attracted a large number of scholars for further studies [16], in which a shell/tube configuration was designed with the shell side (absorbent) parallel to the fiber cells (CO_2_). The influence of CO_2_ absorption efficiency based on physical absorption was carried out in hollow-fiber membrane contactors theoretically and experimentally [17]. Many researchers investigated a high effective MEA absorbent solution of absorbing CO_2_ [18,19], which has been commercialized for many decades with various amines and mixed amines [9,20] used to enhance CO_2_ capture efficiency and reduce regeneration cost [21]. Rongwong et al. [22] provided a better understanding of the CO_2_ removal using MEA absorbent in membrane gas/liquid absorption operations. The current chemical absorption by amines absorbent was confirmed as the most advanced separation technology for CO_2_ absorption [18], and the alkanolamine-based CO_2_ absorption processes have been used commercially. Faiz and Al-Marzouqi [23] developed the mathematical model for the CO_2_ absorption using MEA from natural gas at high pressures, and process intensifications for CO_2_ absorption processes have been investigated successfully by selecting the various membrane materials [24]. The membrane absorption efficiency depending on the distribution coefficient was investigated with the properties of MEA absorbents [25] and the selective membrane materials [24]. Some durable and reusable materials for CO_2_ absorption were proved by Lin et al. [26]. The hybrid silica aerogel and highly porous PVDF/siloxane nanofibrous membranes were combined to enhance the CO_2_ absorption efficiency [27]. The mass transfer performance on the shell side in hollow-fiber membrane modules were examined experimentally [28,29] and reviewed by Lipnizki and Field [30]. The effects of fiber spacing and flow distribution on the device performance were examined to vary significantly [31,32]. The mass-balance and chemical reaction equations were derived to demonstrate the mechanisms of CO_2_ absorption in the hydrophobic porous membrane contactor [20]. The one-dimensional steady-state modeling equation was based on a diffusion–reaction model by considering both chemical absorption and separation technique simultaneously [33]. The CO_2_ absorption flux was obtained under various operational conditions by using amines as absorbents with occurring reactions [34]. In addition, the analytical and experimental studies for shell side mass transfer with fluid flowing axially between fiber cells were investigated by Zheng et al. [35].

In the present work, the theoretical model and experimental work were performed to investigate the CO_2_ absorption in the MEA absorbent using a ceramic hollow-fiber membrane gas/liquid absorption module [36], with gas and liquid flow rates regulated independently. The hollow-fiber precursors fabricated by spinning alumina slurry comprised of alumina powders were used as the main ceramic hollow-fiber membrane materials to validate the theoretical predictions under an ordered fiber arrangement. The theoretical predictions show that the effect of the inlet CO_2_ concentration in the CO_2_/N_2_ feed stream plays an important role in the absorption efficiency. The influences of operating and design parameters, such as packing density (φ), inlet CO_2_ concentration, gas mass flow rate, and absorbent volumetric flow rate on the absorption rate enhancement, are also delineated.

## 2. Theoretical Formulations

A fiber cell model with the imaginary free surface, known the Happel’s free surface [37], was developed [35] to describe the shell side mass transfer characteristics between the shell side with one fiber in each cell of the hollow-fiber module. The Happel’s free surface model was established with the following assumptions: (a) uniformly packed; (b) no friction on the shell side; (c) neglecting the ceramic membrane thickness as compared with the hollow fiber radius; and(d) ignoring the velocity profile across the module radius direction. The radius of fiber cell and free surface are $r_{0}\;$ and $\;r_{f}$, respectively, as shown in Figure 1, being simplified into a circular-tube module. Three regimes considered for modeling CO_2_ absorption in hollow-fiber membrane contactors are shown in Figure 2 in which $\;r_{f}\;\;$ is the free surface radius defined as: (1)$$r_{f}=\varphi^{-0.5}r_{o}$$

$r_{o}$ is the fiber outside radius, φ is the packing density of the hollow fiber module, $r_{f}=r_{o}+D_{f}$, and $\varphi =\frac{\pi r_{o}^{2}}{\;2\sqrt{3}{\left(r_{o}+D_{f}\right)}^{2}}$.

Three mass transfer resistances are built up across the membrane between the bulk flows and membrane in series, as illustrated in Figure 2. The overall mass transfer regions include (1) CO_2_ transfers into the membrane surface from the fiber cell by convection; (2) CO_2_ diffuses by Knudsen diffusion and molecular diffusion through the membrane pores; (3) CO_2_ transfers into liquid side reaching the membrane/liquid interface by convection; and (4) CO_2_ reacted by MEA absorbent. The CO_2_ concentration on the membrane/MEA absorbent interface was determined by the dimensionless Henry’s law constant $H_{c}=\frac{C_{2\left(\ell \right)}}{C_{2\left(g\right)}}=0.73\;$ [20]. In addition, the resistance is controlled by a convective mass transfer that depends on the boundary layer of the MEA absorbent side due to the fast reaction. The mass transfer balance equations were derived for each transfer regime under steady-state operation. The schematic diagram of concentration boundary layers and the CO_2_ concentration variation from the CO_2_/N_2_ feed stream to the MEA absorbent side through the membrane are illustrated in Figure 3.

The mass transfer in the membrane was evaluated by a membrane permeation coefficient ($c_{m}$) [38,39] considering both Knudsen diffusion and molecular diffusion [40], the tortuosity (τ=1/ε) [41], and the trans-membrane saturation partial pressure differences (ΔP) of CO_2_ [42]. The reduced equilibrium constant $K_{ex}^{’}$ is derived to fit in the modeling equation as:(2)$$K_{ex}^{’}=K_{ex}\left[\mathrm{MEA}]/[H^{+}\right]\;$$
in which the equilibrium constant $K_{ex}=\left[{\mathrm{MEACOO}}^{-}\right]\left[H^{+}\right]/\left\{\left[{\mathrm{CO}}_{2}\right]\left[\mathrm{MEA}\right]\right\}=1.25\times 10^{-5}$ at T=298 K [43] for the CO_2_ absorbed in the aqueous MEA absorbent, and can be expressed as follows:(3)$${\mathrm{CO}}_{2}+\mathrm{MEA}↔{\mathrm{MEACOO}\;}^{-}+H^{+}$$

Applications of the dusty gas model [44] to the mass transfer flux in each transport regimes are depicted in Equations (4)–(6), especially in the membrane [40], which was obtained with respect to the concentration driving-force gradient as follows:(4)$$\omega_{g}=k_{a}\left(C_{a\left(g\right)}-C_{1\left(g\right)}\right)$$
(5)$${\left.\omega_{m}=c_{m}\left(P_{1}-P_{2}\right)=c_{m}\frac{dP}{dC}\right]}_{C_{mean}}\left(C_{1}-C_{2\left(g\right)}\right)=c_{m}RT\left(C_{1}-\frac{K_{ex}^{’}C_{2\left(\ell \right)}}{H_{c}}\right)=K_{m}\left(C_{1}-\frac{K_{ex}^{’}C_{2\left(\ell \right)}}{H_{c}}\right)$$
(6)$$\omega_{\ell }=K_{b}\left(\frac{K_{ex}^{’}C_{2\left(\ell \right)}}{H_{c}}-\frac{C_{b\left(\ell \right)}}{H_{c}}\right)$$
where
$$c_{m}={\left(\frac{1}{c_{K}}+\frac{1}{c_{M}}\right)}^{-1}={\left\{{\left[1.064\frac{\varepsilon r_{m}}{\tau \delta_{m}}{\left(\frac{M_{w}}{RT_{m}}\right)}^{1/2}\right]}^{-1}+{\left[\frac{1}{{\left|Y_{m}\right|}_{ln}\frac{D_{m}\varepsilon }{\delta_{m}\tau }\frac{M_{w}}{RT_{m}}}\right]}^{-1}\right\}}^{-1}$$

The amount of mass fluxes from the gas feed stream, transferring through the membrane and then being absorbed into the MEA absorbent stream are all equal by the conservation of mass flux as:(7)$$\omega_{g}=\omega_{m}=\omega_{\ell }$$

The CO_2_ concentrations on the membrane surfaces of both gas and liquid sides can be related in terms of the CO_2_ concentrations of both the bulk gas and liquid streams, with the aid of continuity of mass flux expressed in Equation (7) by Equations (8) and (9), respectively
(8)$$C_{a\left(g\right)}=C_{1\left(g\right)}+\frac{K_{m}}{k_{a}}\left(C_{1\left(g\right)}-\frac{K_{ex}^{’}C_{2\left(\ell \right)}}{H_{c}}\right)$$
(9)$$\frac{C_{b\left(\ell \right)}}{H_{c}}=\frac{K_{ex}^{’}C_{2\left(\ell \right)}}{H_{c}}-\frac{K_{m}}{k_{b}}\left(C_{1\left(g\right)}-\frac{K_{ex}^{’}C_{2\left(\ell \right)}}{H_{c}}\right)$$

Subtracting Equation (8) from Equation (9), one can obtain Equation (10), which can be used to define a concentration polarization coefficient $\gamma_{m}$ [45], exactly the ratio of the bulk concentrations gradient to the membrane surface concentrations gradient, as defined in Equation (11)
(10)$$C_{a\left(g\right)}-\frac{C_{b\left(\ell \right)}}{H_{c}}=\left(C_{1\left(g\right)}-\frac{K_{ex}^{’}C_{2\left(\ell \right)}}{H_{c}}\right)\;\left(1+\frac{k_{m}}{k_{a}}+\frac{k_{m}}{k_{b}}\right)$$
(11)$$\gamma_{m}=\frac{\left(C_{1\left(g\right)}-\frac{K_{ex}^{’}C_{2\left(\ell \right)}}{H_{c}}\right)}{\left(C_{a\left(g\right)}-\frac{C_{b\left(\ell \right)}}{H_{c}}\right)}=\frac{k_{a}k_{b}}{k_{a}k_{b}+k_{m}k_{a}+k_{m}k_{b}}$$

The CO_2_/MEA membrane absorption module configuration includes two separated channels under cocurrent-flow and countercurrent-flow operations, respectively, as shown in Figure 4.

The mass balances of CO_2_/N_2_ feed stream and MEA absorbent stream were calculated within a finite system element, respectively:(12)$$\frac{dC_{a}}{dz}=-\frac{2\pi r_{i}}{q_{a}}\omega_{m}=-\frac{2\pi r_{i}}{q_{a}}\left[k_{m}\gamma_{m}\left(C_{a}-\frac{C_{b}}{H_{c}}\right)\right]$$
(13)$$\frac{dC_{b}}{dz}=\frac{-k_{CO_{2}}C_{b}\pi \left(r_{f}^{2}-r_{o}^{2}\right)}{q_{b}}+\frac{2\pi r_{i}}{q_{b}}\omega_{m}=\frac{-k_{CO_{2}}C_{b}\pi \left(r_{f}^{2}-r_{o}^{2}\right)}{q_{b}}+\frac{2\pi r_{i}}{q_{b}}\left[k_{m}\gamma_{m}\left(C_{a}-\frac{C_{b}}{H_{c}}\right)\right],\;cocurrent-flow operations$$
(14)$$\frac{dC_{b}}{dz}=\frac{k_{CO_{2}}C_{b}\pi \left(r_{f}^{2}-r_{o}^{2}\right)}{q_{b}}-\frac{2\pi r_{i}}{q_{b}}\omega_{m}=\frac{k_{CO_{2}}C_{b}\pi \left(r_{f}^{2}-r_{o}^{2}\right)}{q_{b}}-\frac{2\pi r_{i}}{q_{b}}\left[k_{m}\gamma_{m}\left(C_{a}-\frac{C_{b}}{H_{c}}\right)\right],\;countercurrent-flow operations$$

## 3. Numerical Solutions

The mass balances of Equations (12)–(14) were calculated for CO_2_ gas feed and MEA absorbent with *z*-coordinate along the flowing direction under the cocurrent-flow and countercurrent-flow operations, respectively. Thus, the CO_2_ concentrations in both the bulk streams and membrane surfaces along the module’s length were solved by using the 4th-order Runge–Kutta method. Hence, the CO_2_ absorption flux was obtained. Comparisons were drawn for the CO_2_ absorption efficiency of the module with inserting *N* = 7 and 19 fiber cells under both cocurrent- and countercurrent-flow operations, as shown in Figure 5.

The simultaneous ordinary equations of Equations (12) and (13) for cocurrent-flow operation and Equations (12) and (14) for countercurrent-flow operation in Figure 4a,b, respectively, were solved with the use of the convective mass transfer coefficients. The experimental CO_2_ absorption flux $\omega_{exp}\;$ was used to calculate the convective mass transfer coefficients $k_{b}$ in the MEA feed phase and validated by the CO_2_ absorption flux $\omega_{cal}$ when the iterative procedure reached the convergence tolerance, as shown in Figure 6a. Then, both bulk concentration distributions as well as the CO_2_ absorption flux were calculated numerically in Figure 6b by following the 4th-order Runge–Kutta scheme. An additional guess of CO_2_ concentration at inlet of MEA absorbent feed stream $C_{b,j=n_{step}\;}=0$ was required to be specified in advance in Figure 5b to apply shooting strategy for countercurrent-flow operation. Meanwhile, the absorption rate was defined as:(15)$${\dot{N}}_{j}=\omega_{j}\times \left(2\pi r_{o}L\right)\times N,\;j=single-fiber\;\mathrm{cell}\;(N=1),\;multiple-fiber\;\mathrm{cells}\;(N=7.19)$$

## 4. Enhancement Factor

Multiple-fiber cells were embedded in the MEA feed stream in the hollow-fiber membrane contactor instead of using a single-fiber cell module. The extent of absorption flux increment is incorporated into an enhancement factor [46]. It is also the mass transfer enhancement factor, $\alpha^{E}$, the ratio of the Sherwood number of the module with embedding multiple-fiber cells to that of the module embedding single-fiber cell. The mass transfer enhancement factor $\alpha^{E}$ depending on various fiber cells, packing density (φ), and flow patterns were correlated to demonstrate the augmented mass transfer coefficients in the gas/liquid membrane contactors. The common correlation [47] was used for the membrane contactor under laminar flow as follows:(16)Shlam=0.023Re0.8Sc0.33

The enhancement factor for the mass transfer coefficient can be defined for membrane gas/liquid contactors using embedding multiple-fiber cells instead of single-fiber cell as below:(17)$$Sh^{E}=\frac{k_{b}d_{h,MEA}}{D_{b}}=\alpha^{E}Sh_{lam}$$

The Sherwood number of embedding multiple-fiber cells can be incorporated into four dimensionless groups using Buckingham’s π theorem [48]:(18)$$Sh^{E}=f\left(\frac{L_{b}}{d_{h,MEA}},Re,Sc\right)$$
where $L_{b}$ is the total length of fiber cells inserted, while $d_{h,MEA}$ is the hydraulic diameter in MEA absorbent stream.

The absorption flux enhancement at the expense of the power consumption increment due to friction losses of a gas/liquid membrane contactor with various packing density can be determined using the Fanning friction factor $f_{F}$ [49], including both the CO_2_/N_2_ and MEA sides as:(19)$$H_{i}=q_{a}\;\rho_{CO_{2}/N_{2}}\;\ell w_{f,CO_{2}/N_{2}}+q_{b}\;\rho_{MEA}\;\ell w_{f,MEA\;}$$
(20)$$\ell w_{f,j}=\frac{2f_{F,j}{\overset{-}{v}}_{j}^{2}L}{d_{h,i}},\;j={\mathrm{CO}}_{2}/N_{2},\;\mathrm{MEA}$$
where
$$Re_{CO_{2}/N_{2}}=\frac{\rho_{CO_{2}/N_{2}}{\overset{-}{v}}_{CO_{2}/N_{2}}d_{h,CO_{2}/N_{2}}}{\mu_{CO_{2}/N_{2}}},\;Re_{MEA}=\frac{\rho_{MEA}\;{\overset{-}{v}}_{MEA}\;d_{h.MEA}}{\mu_{MEA}}$$
$${\overset{-}{v}}_{CO_{2}/N_{2}}=\frac{q_{a}}{\pi Nr_{i}^{2}},\;{\overset{-}{v}}_{MEA}=\frac{q_{b}}{\pi \left(r_{s}^{2}-Nr_{o}^{2}\right)},\;d_{h,CO_{2}/N_{2}}=2r_{i},\;d_{h,MEA}=\frac{4\pi \left(r_{s}^{2}-Nr_{o}^{2}\right)}{2\pi \left(r_{s}+Nr_{o}\right)}$$

The relative extents $I_{E}\;$ and $I_{P}\;$ of absorption rate enhancement and power consumption increment, respectively, were illustrated by calculating the percentage increase in the module with inserting multiple-fiber cells on the basis of the module of single-fiber cell as:(21)$$I_{E}=\frac{\omega_{multi}-\omega_{single}}{\omega_{single}}\times 100\%$$
(22)$$I_{P}=\frac{H_{multiple}-H_{single}}{H_{single}}\times 100\%$$

## 5. Experimental Runs

### 5.1. Apparatus and Procedure

The operating and designing parameters include the gas feed volumetric flow rate ($q_{a\;}=3.33{\;\mathrm{cm}}^{3}/s$), liquid absorbent volumetric flow rate ($q_{b\;}=5.0,\;6.67.\;8.33,\;10.0{\;\mathrm{cm}}^{3}/s)$, CO_2_ inlet concentrations (30%, 35%, and 40%), MEA absorbent solution (30 wt%, $5.0\times 10^{-3}{\;\mathrm{mol}/\mathrm{cm}}^{3}$), membrane contactor module ($r_{s}\;$= 0.0075 m, $r_{i}\;$= 0.0004 m, $r_{o}\;$= 0.00065 m, L = 0.17 m, and *N =* 1, 7, and 19), permeability of membrane (ε=0.55), nominal pore size ($r_{m}=0.2\;\mu m$), membrane thickness ($\delta_{m}=250\;\mu m$), solute diffusivity both in gas feed and liquid absorbent ($D_{a\;}$ and $D_{b}$, respectively), and Henry’s law constant ($H_{c}{}_{\;}$). The inorganic hydrophobic membrane is used in the experiments for its superior chemical resistance and thermal stability. The CO_2_ and N_2_ gas mixture was introduced from the well gas mixing tank (EW-06065-02, Cole Parmer Company, Vernon Hills, IL, USA) to flow into the tube side and was regulated by using the mass flow controller (N12031501PC-540, Protec, Brooks Instrument, Hatfield, PA, USA), while the MEA absorbent solution passed through the shell side. The CO_2_ concentrations at the inlet and outlet streams in the experimental runs were collected and measured by using the gas chromatography (Model HY 3000 Chromatograp, China Corporation, New Taipei, Taiwan) to calculate the absorption efficiency. Figure 7a,b illustrate the schematic representations of the hollow-fiber gas/liquid membrane contactor systems for cocurrent- and countercurrent-flow operations, respectively. Duplicate runs were performed under identical operating conditions to ensure reproducibility. Comparisons of the experimental runs and the mathematical predictions were also provided.

### 5.2. Chemicals and Materials

The inorganic hydrophobic fiber–cell membrane [50], with the inner and outer radius of $r_{i}=0.0004\;m$ and $r_{o}=0.00065\;m$, respectively, was used in the experiments, which was prepared in a combined dry–wet spinning alumina slurry comprising alumina powders and a non-solvent deionized water (DI) for phase inversion, followed by a sintering process [51] to prepare the alumina hollow-fiber membranes. The hollow-fiber precursors were fabricated by spinning alumina slurry, including main ceramic materials (alumina powders: 0.7 µm, α- Al_2_O_3_, Alfa Aesar, Haverhill, MA, USA, 99.9% metal basis), solvent (N-Methyl-2-pyrrolidone: NMP, TEDIA, Echo Chemical, Miaoli, Taiwan, purity >99%), binder (polyethersulfone: PES, Veradel A-301, SOLVAY, Trump Chemical, New Taipei, Taiwan, amber color), and dispersant (polyethyleneglycol 30-dipolyhydroxystearate: Arlacel P135, Croda Taiwan, Taiwan, molecular weight: 5000 g mol^−1^). The ceramic hollow-fiber membrane modules were fabricated with various packing densities by encapsulating different numbers of fibers, in which the pinch clamps were sealed at both ends of the tube side using thermoset epoxy, as shown in Figure 5. The packing densities of the hollow-fiber membrane modules were φ  = 0.006, 0.06, and 0.17, with the number of fiber cell of *N* = 1, 7, and 19, respectively.

The membrane surface wettability can be characterized by water contact angle tests. The water contact angles are shown in Figure 8 for the ceramic membranes, which were fabricated specifically for this experiment. Ceramic membrane modification was conducted by mixing 1H,1H,2H,2H-Perfluorooctyltriethoxysilane (FAS-13) and n-Hexane as a grafting agent. The fabricated membranes presented different surface wettability in the range of 139–143° (water contact angle of 141.2 ± 2.0°). On the other hand, the surface hydrophobicity of the hydrophobic ceramic membrane was examined and confirmed.

## 6. Results and Discussions

### 6.1. CO_2_ Absorption Rate Enhancement

This study measured experimentally and predicted theoretically the effects on CO_2_ absorption rate, say $\dot{N}$, for both cocurrent- and countercurrent-flow operations, as depicted in Figure 9. The overall CO_2_ absorption rate was calculated by multiplying the absorption flux by both the number of fiber cells and surface area of each fiber cell. As expected, the increase of MEA feed flow rate, inlet feed CO_2_ concentration, and more fiber cells resulted in a higher absorption rate. The results showed that the CO_2_ absorption rate for the hollow-fiber membrane module increases with embedding more fiber cells in both cocurrent- and countercurrent-flow operations. In general, the module has higher CO_2_ transporting flux through the ceramic hollow-fiber membrane in countercurrent-flow operations than that in cocurrent-flow operations. A larger concentration gradient is accomplished between CO_2_/N_2_ and MEA absorbent in countercurrent-flow operations compared with cocurrent-flow operations, which comes with a higher device performance on CO_2_ absorption rate. Generally, embedding more fiber cells into the shell tube shows a significant influence to increase the absorption rate in the hollow-fiber membrane contactor module.

The absorption flux in the device with embedding various fiber cells are presented graphically for *N* = 1, *N* = 7, and *N* = 19, respectively, as delineated in Figure 10, with the number of fiber cells, inlet feed CO_2_ concentration, and flow pattern as parameters under both cocurrent- and countercurrent-flow operations. The increase of both MEA feed flow rate and the number of fiber cells yielded a higher absorption rate, but the absorption fluxes decreased with the number of fiber cells, as seen in Figure 10.

A relative increment of CO_2_ absorption rate enhancement $I_{E}$ was calculated by comparing the absorption rate with the multiple-fiber cells embedded in the hollow-fiber module with that of the single-fiber cell module. The CO_2_ absorption rate and its improvement for the hollow-fiber module with embedding various fiber cells under both cocurrent- and countercurrent-flow operations can be observed in Table 1 and Table 2. The results indicate that the maximum absorption rate improvement up to 965.5% is obtained as compared with that in the single-fiber cell module. Overall, the CO_2_ absorption rate augmented by embedding more fiber cells substantially increases in countercurrent-flow operations than that in cocurrent-flow operations. The theoretical predictions and experimental results of CO_2_ absorption rate with various MEA feed flow rates and inlet feed CO_2_ concentration under cocurrent- and countercurrent-flow operations are demonstrated in Table 1 and Table 2 and Figure 9, respectively. The results show absorption rate improvement increases with inlet feed CO_2_ concentration but decreases with MEA feed flow rate.

### 6.2. Mass Transfer Enhancement Factor

The mass transfer enhancement factor $\alpha^{E}$ in terms of the total length of inserted fiber cells and the hydraulic diameter of MEA absorbent stream in the hollow-fiber membrane contactor was determined in a regression analysis as:(23)$$\alpha^{E}=2.367\ln{\left(\frac{L_{b}}{d_{h,b}}\right)}^{-0.222}$$

The experimental uncertainty for each measurement of the absorption flux $S_{\omega_{i}}$ was calculated directly as referred to the precision index [52] as follows:(24)$$S_{\omega_{i}}={\left\{\overset{N_{exp}}{\underset{i=1}{\sum }}\frac{{\left(\omega_{exp}-\omega_{theo}\right)}^{2}}{N_{exp}-1}\right\}}^{1/2}$$
and the reproducibility of the absorption flux associated with the mean precision index was obtained by:(25)$$S_{{\bar{\omega }}_{i}}=\frac{S_{\omega_{i}}}{\sqrt{N_{exp}}}$$

The mean precision index of the experimental measurements of absorption flux was evaluated for both cocurrent- and countercurrent-flow operations as $1.02\times 10^{-2}\leq S_{{\bar{\omega }}_{i}}\leq 2.50\times 10^{-2}$. The validation between the theoretical predictions and the experimental results was proved by defining the accuracy [52] as follows:(26)$$E=\frac{1}{N_{exp}}\overset{N_{exp}}{\underset{i=1}{\sum }}\frac{\left|\omega_{theo}-\omega_{exp}\right|}{\omega_{exp}}$$
where $\omega_{theo}$ indicates the theoretical prediction, while $N_{exp}$ and $\omega_{exp}$ are the number of experimental measurements and the experimental data, respectively. The average errors of the experimental measurements were determined by Equation (26) for cocurrent- and countercurrent-flow operations of $1.31\times 10^{-2}\leq E\leq 4.35\times 10^{-2}$ and $3.90\times 10^{-3}\leq E\leq 2.43\times 10^{-2}$, respectively, and are shown in Table 1 and Table 2. The measured absorption fluxes were consistent with the theoretical predictions for CO_2_ absorption in aqueous MEA solutions.

### 6.3. Energy Consumption Increment

A percentage increment of power consumption $I_{P}\;$ was evaluated by comparing the module with embedding multiple-fiber cells with that of using single-fiber cell for *N* = 7 and *N* = 19 under two flow patterns, respectively. Considering the flow friction loss caused by embedding more fiber cells in the MEA feed stream, which consumes more energy consumption, known equivalently as the module design’s effectiveness, comparing the ratio of CO_2_ absorption-rate-enhancement-to-power-consumption increment, $I_{E}/I_{P},\;$ was evaluated to examine the economic feasibility. The effect of MEA absorbent flow rate, the number of fiber cells, inlet feed CO_2_ concentration, and flow patterns on $I_{E}/I_{P}$ are presented in Figure 11. The $I_{E}/I_{P}$ values decrease with the MEA absorbent flow rate but increase with inlet feed CO_2_ concentration. The power consumption increment becomes higher at a larger MEA absorbent flow rate accompanied by a larger mass transfer coefficient for the module with embedding more fiber cells. However, the higher value of $I_{E}/I_{P}$ indicates that the higher absorption rate could compensate for the power consumption increment due to a more fiber cells resulting from a higher CO_2_ absorption efficiency. The theoretical results also found that a comparatively higher $I_{E}/I_{P}$ value for countercurrent-flow operations and fewer number of fiber cells were observed in Figure 11, except at the higher inlet feed CO_2_ concentration and MEA absorbent flow rate, say 40% and $q_{b\;}=10.0{\;\mathrm{cm}}^{3}/s$, respectively. Generally, the comparison reveals that the countercurrent-flow operation can more effectively utilize power supply to increase CO_2_ absorption rate improvement than that of the cocurrent-flow operation. Therefore, comparisons on both *N* = 7 and *N* = 19 fiber cells were made on $I_{E}/I_{P}$ to indicate the trend of economic and technical feasibilities where more fiber cells are embedded in the hollow-fiber membrane contactor of this study.

## 7. Conclusions

A ceramic hollow-fiber gas/liquid membrane contactor, using MEA solution as an absorbent to enhance the CO_2_ absorption rate, was investigated theoretically and experimentally. In addition, mathematical equations were developed on the basis of Happel’s free surface model. The theoretical predictions of the CO_2_ absorption rate improvement were calculated and validated by experimental data, which led to the correlated expression of the Sherwood number for the module by embedding multiple-fiber cells. Embedding two types of multiple-fiber cells into the shell side were implemented, *N* = 7 and *N* = 19, and compared with the module inserting a single-fiber cell. Mathematical treatments in obtaining the absorption rate were derived and presented, with various MEA absorbent flow rates, inlet feed CO_2_ concentrations, and both cocurrent- and countercurrent-flow operations as parameters. The CO_2_ absorption rate increased with MEA absorbent flow rate and inlet feed CO_2_ concentration in the ceramic hollow-fiber membrane contactor by embedding more fiber cells into shell side under both cocurrent- and countercurrent-flow operations, where the larger concentration gradient across membrane surfaces of both feed streams and the membrane surface area were achieved. A maximum absorption rate enhancement up to 965.5% was found in the module by embedding *N* = 19 fiber cells, compared with that in the module of inserting a single-fiber cell. The achieved CO_2_ absorption rate was higher for countercurrent-flow operations than for cocurrent-flow operations, in which the CO_2_ absorption rate was driven mainly by the overall CO_2_ concentration gradient along the flowing direction. The results demonstrate its technical feasibility of absorption rate enhancement in the hollow-fiber membrane contactor. Meanwhile, the effect of the packing density, i.e., the number of fiber cells embedded, on the absorption rate enhancement and increment in power consumption were delineated from an economic perspective. The economic consideration of $I_{E}/I_{P}$ for the absorption-rate-enhancement-to-power-consumption increment indicated that the higher value of $I_{E}/I_{P}$ was achieved for the power utilization’s effectiveness in augmenting CO_2_ absorption rate in this system, where the module with embedding more fiber cells was operated under both the higher MEA absorbent flow rate and inlet feed CO_2_ concentration.

In this paper, both the CO_2_ absorption rate and power utilization effectiveness were examined by implementing various fiber cells in the ceramic hollow-fiber membrane contactor. The alternative absorbent, the membrane material, and the packing density require further investigation on the economic consideration of the ceramic hollow-fiber membrane contactor.

## Figures and Tables

**Figure 1 membranes-13-00249-f001:**
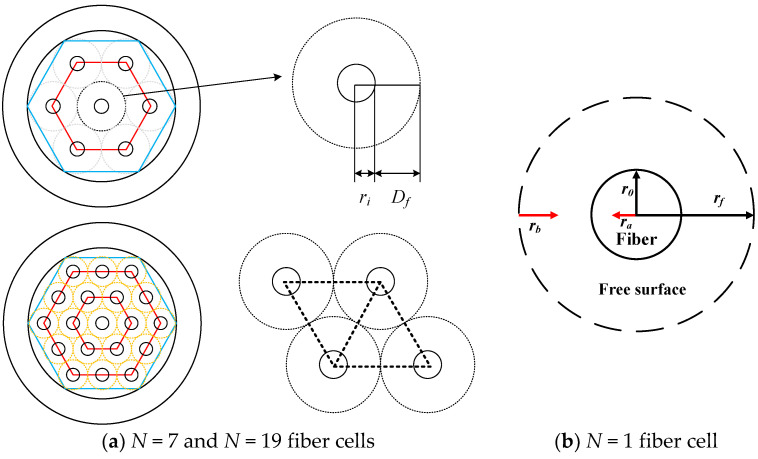
A scheme for the Happel’s free surface model with various fiber cells.

**Figure 2 membranes-13-00249-f002:**
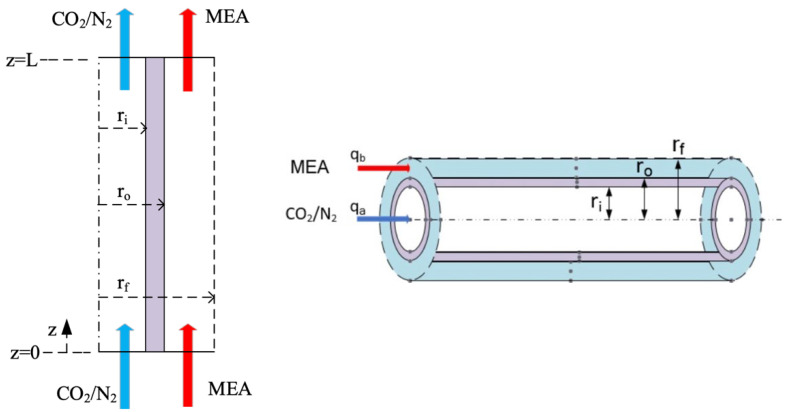
Three regions for modeling CO_2_ absorption in hollow-fiber membrane contactors.

**Figure 3 membranes-13-00249-f003:**
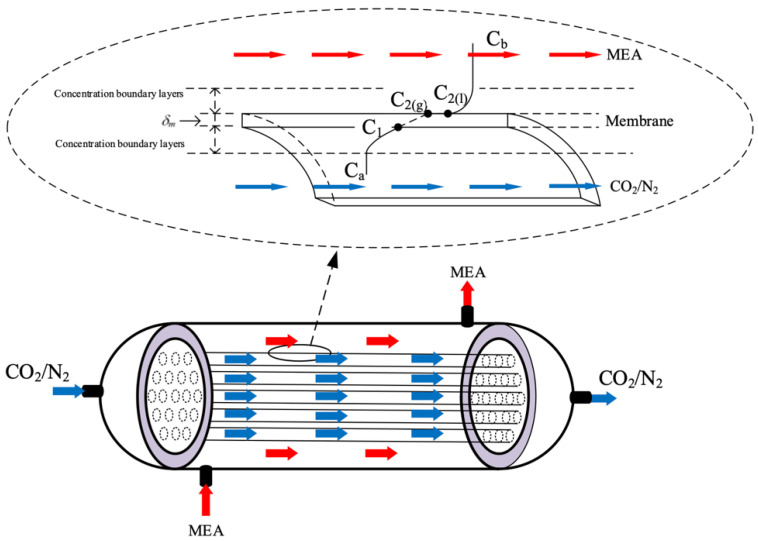
Schematic diagram of concentration boundary layers in a membrane contactor (Both red and blue arrows indicate the flow directions of MEA feed stream and CO_2_/N_2_ feed stream, respectively).

**Figure 4 membranes-13-00249-f004:**
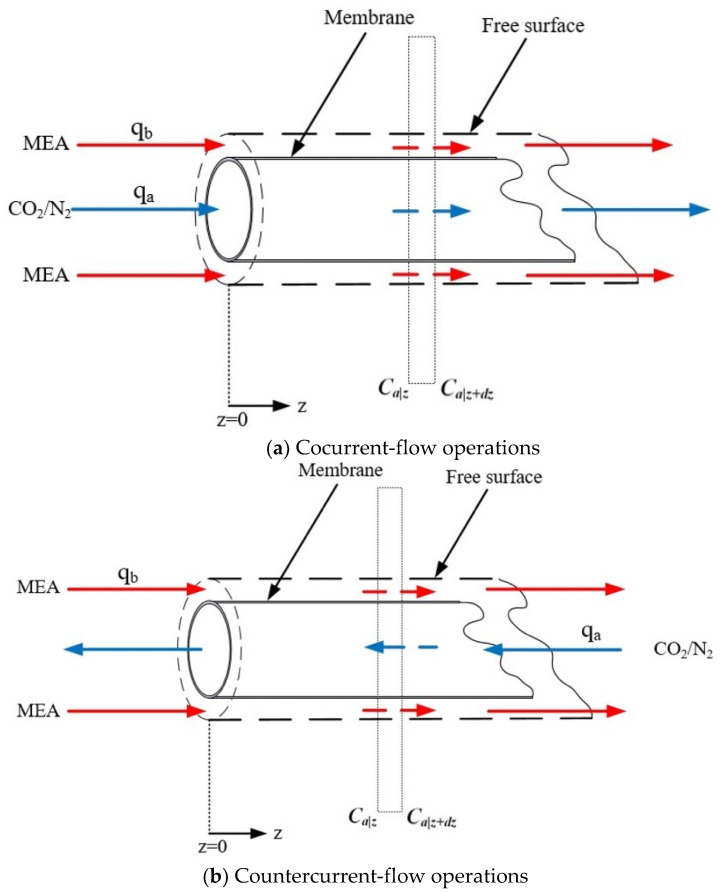
Schematic representation of CO_2_ absorption by MEA for cocurrent- and countercurrent-flow operations in hollow-fiber membrane gas/liquid contactors.

**Figure 5 membranes-13-00249-f005:**
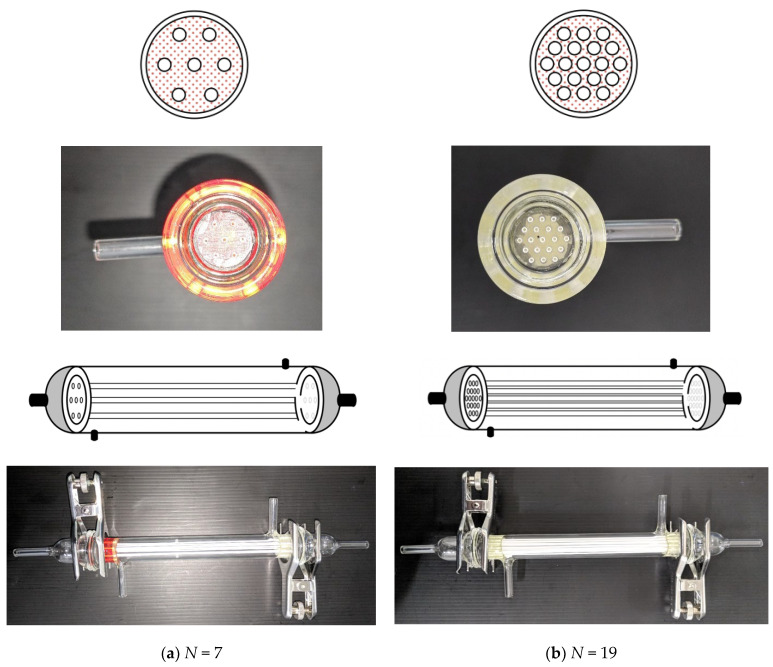
Details of the module configuration with the pinch caps at both ends.

**Figure 6 membranes-13-00249-f006:**
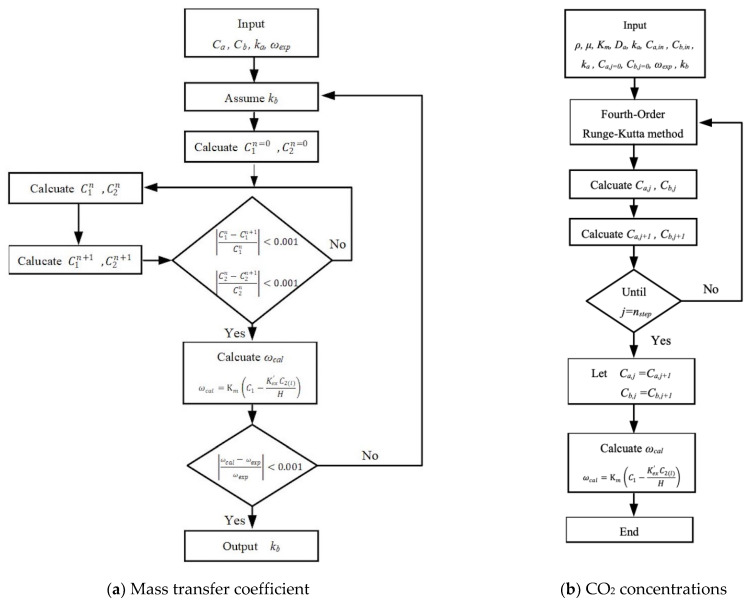
Calculation flow chart for determining the diffusion coefficient and CO_2_ concentrations in gas and liquid phases under cocurrent-flow operations.

**Figure 7 membranes-13-00249-f007:**
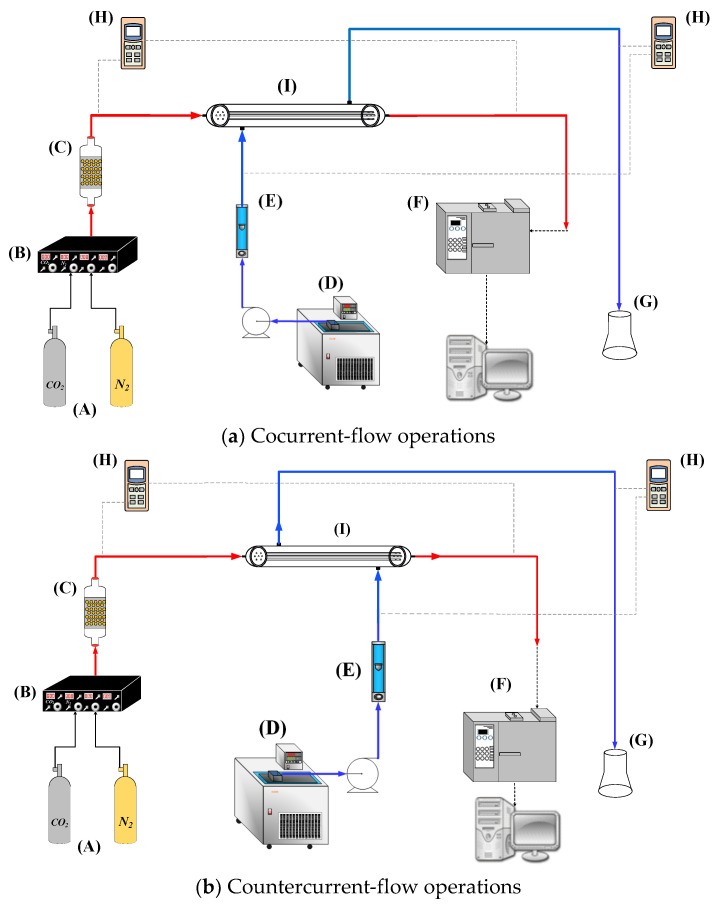
Experimental setup for hollow-fiber gas/liquid membrane contactors: (A) gas cylinder; (B) mass flow controller; (C) gas mixing tank; (D) thermostatic tank; (E) flow meter; (F) chromatography; (G) Erlenmeyer flask; (H) temperature indicator; and (I) ceramic hollow-fiber module.

**Figure 8 membranes-13-00249-f008:**
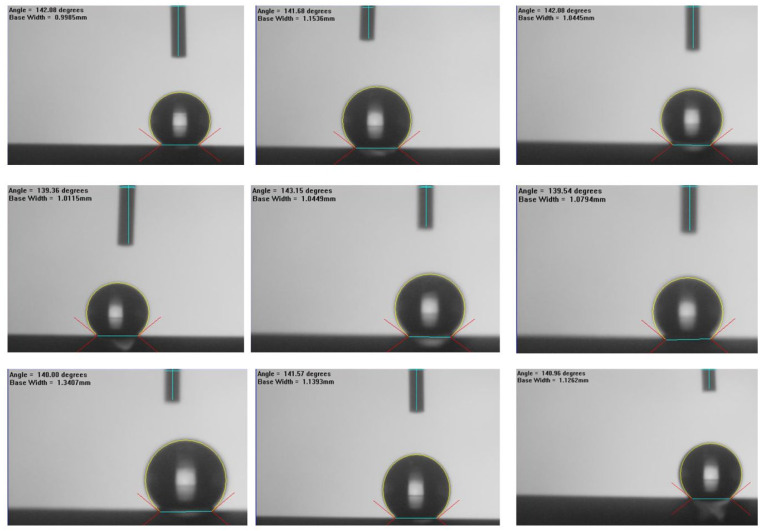
Water contact angles of the fabricated ceramic hollow-fiber membranes.

**Figure 9 membranes-13-00249-f009:**
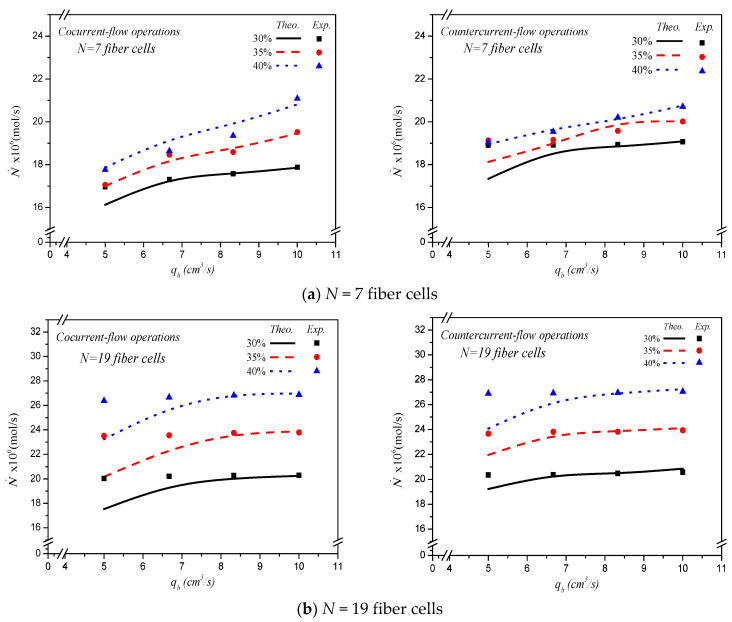
Effects of MEA flow rate and various fiber cells on CO_2_ absorption flux.

**Figure 10 membranes-13-00249-f010:**
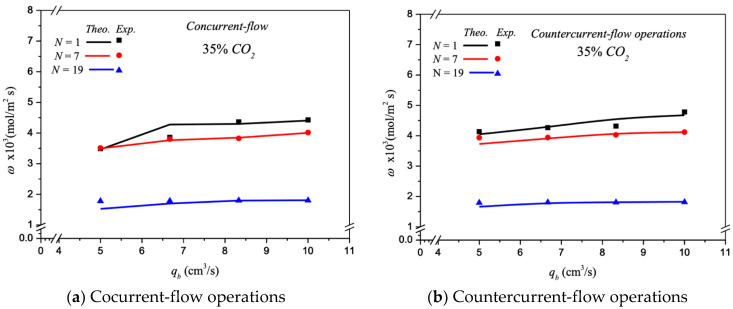
Effects of MEA flow rate and various fiber cells on CO_2_ absorption flux.

**Figure 11 membranes-13-00249-f011:**
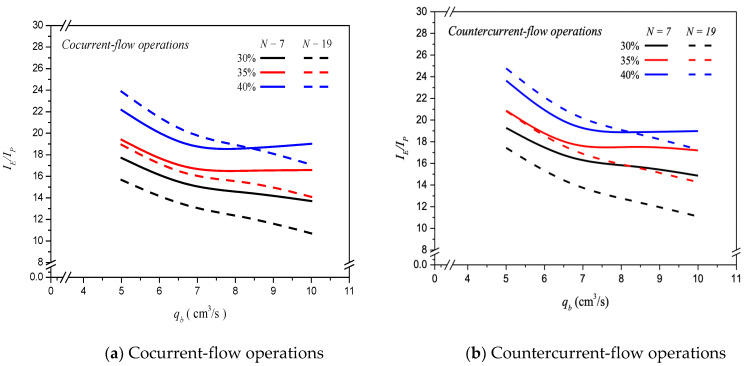
Effects of MEA flow rate, various fiber cells, and feed CO_2_ concentration on $I_{E}/I_{P}$.

**Table 1 membranes-13-00249-t001:** Effects of MEA flow rate and various fiber cells on *I_E_* for cocurrent-flow operations.

$C_{in}(\%)$	$q_{b}\times 10^{6}\;(m^{3}\;s{-}^{1})$	Single-Fiber Cell		Multiple-Fiber Cells					
				*N* = 7			*N* = 19		
		${\dot{N}}_{theo}\times 10^{5}\;(\mathrm{mol}\;s{-}^{1})$	E (%)	${\dot{N}}_{theo}\times 10^{5}\;(\mathrm{mol}\;s{-}^{1})$	E (%)	$I_{E}$	${\dot{N}}_{theo}\times 10^{5}\;(\mathrm{mol}\;s{-}^{1})$	E (%)	$I_{E}$
30	5.00	2.47	4.84	16.13	4.94	553.1	19.23	5.53	609.9
	6.67	3.05	1.55	17.41	0.65	471.1	20.43	0.29	540.1
	8.33	3.12	3.20	17.57	0.03	463.3	20.44	0.22	543.8
	10.0	3.29	9.42	17.86	0.1	442.8	20.87	1.42	515.4
35	5.00	2.41	0.61	17.00	0.34	606.0	21.96	7.19	737.7
	6.67	2.97	10.9	18.31	0.82	516.2	23.69	0.50	655.6
	8.33	2.99	1.44	18.72	0.71	527.1	23.86	0.19	694.3
	10.0	3.06	0.41	19.46	0.29	536.0	24.11	0.74	680.1
40	5.00	2.26	1.53	17.87	0.59	692.3	24.08	10.5	930.5
	6.67	2.83	0.96	19.25	3.28	580.3	26.42	1.88	815.0
	8.33	2.87	0.19	19.86	2.63	591.5	26.94	0.05	838.3
	10.0	2.91	0.86	20.81	1.31	614.6	27.24	0.69	826.1

**Table 2 membranes-13-00249-t002:** Effects of MEA flow rate and various fiber cells on *I_E_* for countercurrent-flow operations.

$C_{in}(\%)$	$q_{b}\times 10^{6}\;(m^{3}\;s{-}^{1})$	Single-Fiber Cell		Multiple-Fiber Cells					
				*N* = 7			*N* = 19		
		${\dot{N}}_{theo}\times 10^{5}\;(\mathrm{mol}\;s{-}^{1})$	E (%)	${\dot{N}}_{theo}\times 10^{5}\;(\mathrm{mol}\;s{-}^{1})$	E (%)	$I_{E}$	${\dot{N}}_{theo}\times 10^{5}\;(\mathrm{mol}\;s{-}^{1})$	E (%)	$I_{E}$
30	5.00	2.47	4.84	17.34	8.27	602.0	19.23	5.53	678.6
	6.67	3.05	1.55	18.51	2.13	506.9	20.43	0.29	569.9
	8.33	3.12	3.20	18.85	0.47	504.2	20.44	0.22	555.1
	10.0	3.29	9.42	19.08	0.09	480.0	20.87	1.42	534.4
35	5.00	2.41	0.61	18.13	5.23	652.3	21.96	7.19	811.2
	6.67	2.97	10.9	18.99	0.84	539.4	23.69	0.50	697.7
	8.33	2.99	1.44	19.86	1.47	564.2	23.86	0.19	698.0
	10.0	3.06	0.41	20.06	0.02	555.6	24.11	0.74	687.9
40	5.00	2.26	1.53	18.95	0.46	738.5	24.08	10.5	965.5
	6.67	2.83	0.96	19.63	0.50	593.7	26.42	1.88	833.6
	8.33	2.87	0.19	20.13	0.36	601.4	26.94	0.05	838.7
	10.0	2.91	0.86	20.76	0.23	613.4	27.24	0.69	836.1

## Data Availability

Data are contained within the article.

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
