# Peer review of "Investigation on the Performance of CO2 Absorption in Ceramic Hollow-Fiber Gas/Liquid Membrane Contactors"

_membranes, 2023, doi:10.3390/membranes13020249_

Round 1
Reviewer 1 Report
The work is about ceramic hollow fiber membrane contactor for CO2 absorption. The work itself is meaningful. However, the design of experiment is too simple and the analyses of experimental observations and comparison with modeling data are not very solid. The manuscript needs significant improvement.
1. Lines 99-103: "...as known he Happel's free surface". What does "which" in line 101 refer to? The sentence is too long and not clearly organized. The authors may check the whole manuscript for grammatical errors, e.g., "Thus, the mass flux...were obtained, and was depicted in..." (line169) and "The operating and designing parameters were provided in this stud, which include..." (lines 299-305).
2. Figure 1: Which one in Figure 1(a) shows 9 fiber cells?
3. The references of the equations should be clearly presented.
4. Section 5.1: Is the membrane home-made or obtained from any supplier? For the dimension of the ceramic hollow fibers, can the author check and confirm that ri=0.0004m=0.4mm, ro=0.00065m=0.65mm? As the ceramic membrane is home-made, details of the fabrication should be presented. If not, a reference disclosing the fabrication details would also help. The authors mentioned that the membranes are hydrophobic. Did the authors conduct any tests (e.g., water contact angle) to examine its hydrophobicity?
5. Figure 7: 1) Is it "cocurrent" or "concurrent"? 2) The length of the contactor is 0.17m, which is a small. In Figure 7, it looks like a big element as compared to other items. The authors might purposely draw in such way to highlight the membrane, but the ratio is too far away from the actual dimensions.
6. Section 6.1: discussed mass-transfer enhancement factor but corresponding tables of data appeared in after Section 6.2. Section 6.1 defined four technical terms and showed the results. However, there were negligible discussions, analyses or elaborations.
7. Lines 344-345: Equation (22) is for the determination of enhancement factor instead of the Sherwood number. Please try to improve the statement.
8. Section 6.2, Figure 8: As the contactor was operated at very slow MEA flow rate, the impact on the CO2 absorption is hardly reflected. As the variable is only flow rate, it may be good if the x-axis is directly shown as flow rate instead of Re. Obviously, a big difference is seen between the experimental CO2 flux and the calculated one. However, there is no discussion here. Figure 9 actually presented the same data of Figure 8. The statement of "embedding more fiber cells into the shell tube shows a significant influence to increase the absorption rate" seems correct. However, it may give different conclusion if the CO2 absorption rate is calculated based on per membrane surface area. This also applies to subsequent statements (e.g., lines 396-399).
Author Response
Dear Reviewer,
Attached file is the revised manuscript entitled, “membranes-2212949, Investigation on the Performance of CO2 Absorption in Ceramic Hollow-Fiber Gas-Liquid Membrane Contactors”, which has been submitted to Membranes(Special Issue: The Latest Achievements and Study Progress of Metal–Organic Framework (MOF) Membrane). It has been carefully revised and edited in which reviewers’ comments and your suggestions are incorporated. All the revisions in the revised manuscript are marked with font color in red. An Itemized Response to the Reviewers’ Comments is also enclosed.
I would like to express our thanks for the valuable comments and inputs from you and this scientific community. We hope that this revised manuscript would meet the Journal’s expectation and be considered for publication onMembranes. Your further consideration is greatly appreciated.
Best regards,
Chii-Dong Ho, Ph.D.
Distinguished Professor
Department of Chemical and Materials Engineering
Tamkang University
Tamsui, Taipei

Reviewer 2 Report
I have checked the manuscript as thoroughly as possible. I could not find any serious flaws in the model they developed and the flux data look also reasonable. The model is applicable for the hollow fiber module design. I recommend major revision for the following reasons.
1) English is very poor.
2) I found the following typo errors, indicating that the manuscript was not carefully prepared.
Fig. 1 (a) N=9
Equation (11) and many other equations H is used for Henry constant.
3) Even though the authors say that the alumina membrane they prepared is hydrophobic, I am not convinced by seeing the materials and the method they used for the preparation of the membrane. The authors should show the contact angle data.
4) The absorption rate N (dot) is given in Fig. 8 and other figures without being defined. Please clarify what it means and how it was obtained.
Author Response

(The authors gave the same response as above.)

Round 2
Reviewer 1 Report
The authors have answered the questions and the manuscript has been significantly improved after the revision.
Author Response
Dear Reviewer,
Attached file is the re-revised manuscript entitled, “membranes-2212949, Investigation on the Performance of CO2 Absorption in Ceramic Hollow-Fiber Gas-Liquid Membrane Contactors”, which has been submitted to Membranes (Special Issue: The Latest Achievements and Study Progress of Metal–Organic Framework (MOF) Membrane). It has been carefully re-revised and edited in which reviewers’ comments and your suggestions are incorporated. All the re-revisions in the re-revised manuscript are marked with font color in red. An Itemized Response to the Reviewers’ Comments is also enclosed.
I would like to express our thanks for the valuable comments and inputs from you and this scientific community. We hope that this re-revised manuscript would meet the Journal’s expectation and be considered for publication onMembranes. Your further consideration is greatly appreciated.
Best regards,
Chii-Dong Ho, Ph.D.
Distinguished Professor
Department of Chemical and Materials Engineering
Tamkang University
Tamsui, Taipei
Taiwan 251

Reviewer 2 Report
The answer to the last question does not make sense.
You have written
N(dot) = Omega x N
Omega has dimension of mol/m2 s, N(dot) has the dimension of mol/s, and N is number. So, the dimension does not match in the equation. Most likely, your equation should be
N(dot)= Omega x (2 pai rL) x N
r is the radius of the fiber. It will be either ri or rf, depending on which radius was used to define your flux. L is the length of the fiber. Together, 2 pai rL is the area of one fiber.
Please think about the equation once more. Examining the dimension match is very important in simulation, because by this way we can find the error more easily.
Author Response

(The authors gave the same response as above.)
